# Neural Responses to a Working Memory Task in Acute Depressed and Remitted Phases in Bipolar Patients

**DOI:** 10.3390/brainsci13050744

**Published:** 2023-04-29

**Authors:** Juliane Kopf, Stefan Glöckner, Heike Althen, Thais Cevada, Martin Schecklmann, Thomas Dresler, Sarah Kittel-Schneider, Andreas Reif

**Affiliations:** 1Department of Psychiatry, Psychosomatic Medicine and Psychotherapy, University Hospital, Goethe University Frankfurt, 60590 Frankfurt am Main, Germany; 2Department of Psychiatry, Psychosomatics and Psychotherapy, University Hospital Wuerzburg, 97080 Wuerzburg, Germany; 3Sport Science Program (PPGCEE), State University of Rio de Janeiro (UERJ), Rio de Janeiro 20550-013, Brazil; 4Department of Psychiatry, Psychosomatics and Psychotherapy, University Hospital of Regensburg, 93053 Regensburg, Germany; 5Department of Psychiatry and Psychotherapy, Tuebingen Center for Mental Health, University of Tuebingen, 72076 Tuebingen, Germany; 6LEAD Graduate School & Research Network, University of Tuebingen, 72072 Tuebingen, Germany

**Keywords:** verbal n-back, fNIRS, prefrontal cortex, cognitive deficits, bipolar disorder, remitted/acute phase

## Abstract

(1) Cognitive impairments such as working memory (WM) deficits are amongst the most common dysfunctions characterizing bipolar disorder (BD) patients, severely contributing to functional impairment. We aimed to investigate WM performance and associated brain activation during the acute phase of BD and to observe changes in the same patients during remission. (2) Frontal brain activation was recorded using functional near-infrared spectroscopy (fNIRS) during n-back task conditions (one-back, two-back and three-back) in BD patients in their acute depressive (*n* = 32) and remitted (*n* = 15) phases as well as in healthy controls (*n* = 30). (3) Comparison of BD patients during their acute phase with controls showed a trend (*p* = 0.08) towards lower dorsolateral prefrontal cortex (dlPFC) activation. In the remitted phase, BD patients showed lower dlPFC and ventrolateral prefrontal cortex (vlPFC) activation (*p* = 0.02) compared to controls. No difference in dlPFC and vlPFC activation between BD patients’ phases was found. (4) Our results showed decreased working memory performance in BD patients during the working memory task in the acute phase of disease. Working memory performance improved in the remitted phase of the disease but was still particularly attenuated for the more demanding conditions.

## 1. Introduction

Bipolar affective disorder (BD) is one of the most debilitating illnesses, affecting about 60 million people worldwide. Thus, mood typically switches between phases of acute depression, mania or hypomania, and euthymia. Moreover, cognitive deficits, such as working memory (WM) impairments, are amongst the most common dysfunctions characterizing BD patients, compromising their daily life skills and contributing to their inability to function well [1]. Executive deficits have been shown to have broad and significant implications in “real life” [2], often demonstrated as severe problems in the control and regulation of emotional behavior [3,4]. Although especially pronounced during the acute phase of the disorder [5], it has also been shown that remitted BD patients still display cognitive deficits [6]. Here, it seems that WM is most notably impaired during the acute phase and remains impaired in remission as well. However, research on the neuronal underpinnings of this behavioral deficit is sparse, and the results are contradictory [7].

WM impairment has been considered an endophenotypic marker of BD [8,9]. The n-back task is a broadly applied and reliable paradigm to investigate WM performance and has been successfully used in fMRI [10] and functional near-infrared spectroscopy (fNIRS) studies [11,12,13]. Furthermore, the usage of the n-back task as one of the most extensively applied paradigms for the assessment of WM in BD patients has been highlighted [2]. A review of the neural underpinnings of WM, as measured with the n-back task, points to a loss of connectivity in BD patients’ prefrontal cortex (PFC) networks as well as patterns of abnormal activation in the dorso-/ventrolateral PFC and the parietal and temporal cortex [9]. Moreover, a negative correlation between n-back performance and the duration of BD illness has been found [14]. 

Although there is a large literature on PFC activation during WM in BD patients [15,16,17,18,19,20,21,22,23,24,25], no previous studies have utilized a longitudinal design and compared the acute and remitted phase in the same sample of BD patients. The few studies that compared the acute and remitted phases of BD patients (in different samples) did not find any difference in PFC activation [18,26]. Compared to controls, BD patients showed attenuated patterns of activity in the PFC across WM performance in the acute [23] and remitted phases [15,19,20,22,25]. In contrast to these findings, other studies report increased activation in the frontopolar PFC [24] and middle frontal gyrus [27] and no difference in dorsolateral PFC (dlPFC) activation [18,21,26,28] when remitted BD patients are compared to controls. To summarize, the majority of studies point to WM impairment with decreased PFC activation in BD patients compared to controls, but these results seem less consistent regarding BD patients’ phase. 

It is noteworthy that, according to our knowledge, no study has been conducted measuring the WM of the same patients across different phases and compared them with matched controls (age, sex, and intelligence). The aim of this study, therefore, was to investigate brain activation patterns during the acute phase and observe changes in the same patients during the remission phase. We applied the n-back task to measure working memory performance and the related PFC activation. We hypothesized that, consistent with earlier research from our group [29] comparing the BD phases and controls, the WM performance would be worse in the acute phase and would improve in the remitted state but would not reach performance levels of an age- and intelligence-matched healthy control group. Against the background of the endophenotypic marker hypothesis of WM deficits in BD, we hypothesized PFC brain activation to be most attenuated in acute phase patients, followed by remitted patients, when compared to controls.

## 2. Materials and Methods

The present sample has been described before [29]. The description can also be found in the Appendix A. The sample was composed of thirty-two BD patients, with both bipolar I and bipolar II subtypes of the disorder in their acute depressive phase, who were recruited from a specialized bipolar ward at the Department of Psychiatry at the University of Wurzburg, and thirty healthy controls. The controls knew about the research through flyers, which were disseminated in the city, and voluntarily appeared. 

BD patients in their acute depressive phase were screened by two trained psychiatrists (AR, SKS) and diagnosed in accordance with both the DSM-IV (since this study was conducted in 2010, DSM-IV was still applicable) and OPCRIT diagnostic system [30]. The difference between the DSM-IV and the OPCRIT system is that one is mainly used in scientific settings and allows for a better compatibility of diagnoses across different study designs. It is an approach to develop a data collection method that can be used across a range of clinical and research settings. Moreover, the Montgomery–Åsberg Depression Rating Scale (MADRS) [31] and the Young Mania Rating Scale (YMRS) were applied to quantify disease severity and to exclude mixed episodes [32], respectively. Medication was recorded to assess possible confounding effects of the results. No patient was medication-free and all had more than one medication as follows: 24 patients had antipsychotic medication, 3 took more than one antipsychotic medication, 18 took lithium, 14 took SSRIs, and 8 had tricyclic anti-depressive medication. Healthy control participants were matched to the sample of acute BD patients regarding age, sex, and performance on the multiple-choice word test (MWT-B) [33], which measures crystallized intelligence. Control participants underwent the Mini International Neuropsychiatric Interview (MINI), German Version 5.0 [34], to exclude current or past axis I mental disorders (classified via the DSM-IV). 

For a follow-up measurement, at least three months later and after reaching remission, 15 of the initial 32 patients achieved the established criteria for remission, assessed by a psychiatrist and confirmed again with MADRS (the score was below six for all patients) and YMRS (no patient scored higher than two as a cut-off) assessments. In the remitted phase, the patients’ medication was reported as follows: one patient was medication-free, four patients had a medication monotherapy, eight patients had antipsychotic medication, one took more than one antipsychotic medication, nine took lithium, six took SSRIs, and six had tricyclic antidepressive medication. The list of medications for BD patients in both phases is provided as Appendix A. 

This study was reviewed and approved by the Ethics Committee of the University of Wurzburg, and all procedures were in accordance with the latest version of the Declaration of Helsinki. All test subjects gave written informed consent after comprehensive explanation of the experimental procedures.

The task is described in detail in Kopf et al. [11]. Briefly, nine letters were presented in a randomized sequence in blocks of fifteen letters, of which three letters were target letters. The blocks contained one-back, two-back, and three-back difficulty levels, named as task conditions. For the one-back task as well as two-back task training, 15 stimuli were presented. Each trial contained three target letters, one of which was a cue letter. For the three-back task, 60 stimuli were presented containing 11 target letters; 2 also functioned as cue letters. Before each block, test subjects were instructed as to which difficulty level would follow next. Each letter was presented for 300 ms, and letters were divided by a blank screen lasting 1700 ms. Each block was 30 s long and was followed by a 30 s resting block. 

Before the experiment, the subjects trained for the three task conditions with different letters. Furthermore, some concerns on n-back’s application might be applied: information must be monitored, updated, and temporally tagged, and responses to intervening stimuli must be inhibited [35].

The basic functionality of functional near-infrared spectroscopy (fNIRS) is described in detail in publications by Hoshi as well as by Obrig and Villringer [36,37]. We used a continuous wave system (ETG-4000 Optical Topography system; Hitachi Medical Co., Tokyo, Japan) with a 52-channel array of optodes covering about 30 × 6 cm of the forehead (interoptode distance = 3cm). Seventeen light emitters (semiconductor lasers) and sixteen photo-detectors (avalanche photodiodes) formed the array, with each detector collecting the reflected near-infrared light of its surrounding emitters. A measuring point of activation (channel) was defined as the region between one emitter and one detector. This probe set was placed on the head according to the standard positions of Fpz (for detector optode 26) and T3/T4 (for emitter optodes 28 and 23) according to the International 10–20 system for EEG electrode placement [38]. The probe set covered both the left and right frontal cortex areas. 

Conducted comparisons were: (1) BD patients in their acute depressive phase (acute; *n* = 32) vs. healthy control subjects (controls; *n* = 30); (2) a subsample of BD patients, who reached remission, in their remitted phase (remitted/R; *n* = 15) vs. healthy control subjects (controls; *n* = 30); and (3) a subsample of BD patients, who reached remission, in their acute depressive phase (remitted/A; *n* = 15) vs. patients in their remitted phase (remitted/R; *n* = 15).

Since patients were assessed in their acute and remitted phases, dependent variable tests were applied. Otherwise, whenever acute BD patients or remitted BD patients were compared with controls, independent variable tests were applied. All conducted statistical tests were two-tailed, and the Kolmogorov–Smirnoff test was conducted first to ensure normal distribution of the data. The significant difference adopted was *p* ≤ 0.05 and values *p* < 0.10 were considered to demonstrate significant differences. Data analysis was conducted using SPSS (IBM Corp. Released 2011. IBM SPSS Statistics for Windows, Version 20.0. Armonk, NY, USA: IBM Corp.).

For the analysis of reaction time effects, only correct responses were considered. To analyze differences in reaction time, three different 3 × 2 (mixed) repeated-measures analyses of variance (ANOVA) were conducted for the factor condition (one-back, two-back, three-back task) and the factor group/phase (acute vs. controls; remitted/R vs. controls; remitted/A vs. remitted/R). Post hoc *t*-tests were conducted, when ANOVA results permitted. 

Since very few errors were made overall, for the analysis of error rates, errors were averaged over all three difficulty levels and then analyzed. The Kolmogorov–Smirnoff test was conducted first to ensure normal distribution of the data. If the assumption of normality was violated, data were analyzed with a Mann–Whitney U test for the independent group analysis and with a Wilcoxon test for the dependent group analysis. If the assumption of normality was adhered to, analyses were performed using independent or dependent *t*-tests. 

For the fNIRS analysis, first, the high-frequency portion of the signal gathered with the ETG-4000 was removed by applying the built-in moving average (MA) filter with a time window of 5 s. To remove slow drifts, a three-element discrete cosine transform basis set was then used on the data. We chose the last five seconds before each block as the baseline period, since the neural response at that time was hypothesized to be near zero. Thereafter, the last 27 s of the 30 s long block were defined as the activation period since, three seconds after the initial trigger, the neural response should already see an increase in activation [39]. 

All data were corrected to control for head motion artifacts based on correlation-based signal improvement [40]. This emphasizes that O2Hb and HHb should be almost perfectly negatively correlated. The outcome provides a corrected signal from the linear combination of O2Hb and HHb parameters. For details, see Cui et al. [40]. The mean of the Cui-corrected O2Hb and HHb concentrations was computed for each segment. Afterwards, the means of the baseline were subtracted from the activation phase. 

We assigned fNIRS channels to specific brain areas [41] and defined regions of interest (ROIs) for both hemispheres. Based on previous work [23,35], we defined the ventrolateral PFC (vlPFC) (Brodman areas 44/45) and the dlPFC (Brodman areas 9/10/46) as ROIs (right vlPFC: 34, 35, 45, 46; left vlPFC: 39, 40, 49,50; right dlPFC: 3,4,13,14, 15, 24, 25; left dlPFC: 7, 8, 17, 18, 19, 28, 29). 

For statistical analyses of the fNIRS data, six separate 2x3x2 ANOVAs were calculated for the activation (mean values) in the two different ROIs (vlPFC, dlPFC; three ANOVAs per ROI). For the comparison of the acute group and control group, as well as the remitted/R group vs. the control group, mixed ANOVAs consisted of the two dependent factors, hemisphere (left, right) and task condition (one-back, two-back, three-back), and the independent factor group (acute vs. controls; remitted/R vs. control). For the comparison of the phases in the remitted group, repeated-measures ANOVAs with the dependent factors of hemisphere (left, right), task condition (one-back, two-back, three-back), and phase (remitted/R vs. remitted/A) were conducted. Subsequently, relevant post hoc *t*-tests were conducted. One patient had to be excluded due to massive artifacts in the NIRS data.

## 3. Results

Details of sociodemographic and descriptive data are described in Table 1. A significant statistical difference in MADRS (t = 5.32; *p* < 0.001) and YMRS (t = 3.85; *p* = 0.004) scores between remitted/A vs. remitted/R was found, as expected. All groups were age-matched: acute vs. control (t = 0.14; *p* = 0.90) and remitted/R vs. control (t = 1.09; *p* = 0.07). Although there was no gender difference between acute BD patients and controls (X^2^ = 2.5; *p* = 0.10), there was a larger sample of female subjects in the healthy group compared with the remitted/R BD group (X^2^ = 4.5; *p* = 0.03).

### 3.1. Behavioral Results

#### 3.1.1. Reaction Time 

All ANOVAs revealed a main effect of task condition (acute vs. control: F (1.64, 98.25) = 46.42, *p* < 0.001; remitted/R vs. control: F (2, 86) = 68.41, *p* < 0.001; and remitted/A vs. remitted/R: F (2, 28) = 33.63, *p* < 0.001) with worsening performance as task condition difficulty increased. Furthermore, there was a main effect of the factor group/phase (acute vs. control: F (1, 60) = 14.95, *p* < 0.001; remitted/R vs. control: F (1, 43) = 4.96, *p* = 0.03; and remitted/A vs. remitted/R: F (1, 14) = 6.85, *p* = 0.02) with longer n-back reaction time for the BD patients compared to controls and for the acute depressed BD patients compared to remitted BD patients. 

ANOVAs revealed no interaction of group/phase*condition: acute vs. control (F(1.64,98.25) = 1.01, *p* = 0.37), remitted/R vs. control (F(2,86) = 2.12, *p* = 0.13), and remitted/A vs. remitted/R (F(2,28) = 0.07, *p* = 0.94). 

Post hoc *t*-tests for the main effects of the condition revealed significant differences in all three conditions for the acute vs. control comparison (one-back: t(48.17) = 3.97, *p* < 0.01; two-back: t(48.69) = 3.84, *p* < 0.01; three-back: t(60) = 2.49, *p* = 0.01). Moreover, the two-back and three-back conditions showed significant differences for the comparison of remitted/R vs. controls (one-back: t(43) = 1.35, *p* = 0.19; two-back: t(43) = 2.05, *p* = 0.04; three-back: t(43) = 2.21, *p* = 0.03). In contrast, only the one-back condition resulted in a significant difference for remitted/A vs. remitted/R (one-back: t(14) = 2.66, *p* = 0.02; two-back: t(14) = 2.03, *p* = 0.06; three-back: t(14) = 1.83, *p* = 0.08). These results indicate a faster reaction time for controls when compared with BD patients and for remitted/R when compared with remitted/A (see Figure 1). The n-back task reaction times, separated by groups/phases, are presented in Figure 1. 

#### 3.1.2. Commission Errors

The comparisons between groups/phases revealed no significant differences for commission error (averaged over all n-back difficulties): acute vs. controls (t(60) = 1.01, *p* = 0.32), remitted/R vs. controls (t(43) = 1.22, *p* = 0.23), and remitted/A vs. remitted/R (t(14) = 1.32, *p* = 0.21)—see Figure 2.

#### 3.1.3. Omission Errors

Comparing the omission errors (averaged over all n-back difficulties) across groups/phases showed significant difference for acute BD patients vs. controls (U = 269.5, *p* = 0.003), with acute BD patients committing a higher number of omission errors than controls. In contrast, there was no significant difference between remitted/R BD patients vs. controls (t(43) = 0.37, *p* = 0.71). Furthermore, there was a trend towards significance between both BD phases (remitted/A vs. remitted/R; t(14) = 1.96, *p* = 0.07), with more errors in the remitted/A phase—see Figure 2.

### 3.2. Imaging Results

#### 3.2.1. dlPFC

Comparison of the activation (mean values) in the ROI dlPFC with ANOVAs revealed a significant main effect of hemisphere with a higher activation in the right dlPFC for acute BD patients vs. controls (F(1, 60) = 5.58, *p* = 0.02) and for remitted/R vs. controls (F(1, 42) = 5.86, *p* = 0.02). Moreover, there was a significant group main effect with controls showing higher dlPFC activation compared to remitted/R patients (F(1, 42) = 5.33, *p* = 0.02) and a marginally significant group effect with higher dlPFC activation for the controls vs. acute BD patients (F(1, 60) = 3.07, *p* = 0.08). Analysis of remitted/A vs. remitted/R showed no main effects (*p* > 0.17). No main effect of condition was found in any ANOVA (*p* > 0.45).

For acute BD patients vs. controls, statistical comparison revealed a significant interaction effect of hemisphere*group (F(1, 60) = 7.23, *p* = 0.01). This indicated higher activation in the right dlPFC for controls compared to acute phase BD patients. Moreover, comparison of remitted/A vs. remitted/R patients showed a trend towards significance for hemisphere*phase (F(1, 13) = 3.99, *p* = 0.06), and no statistically significant interaction hemisphere*group for remitted/R BD patients vs controls (*p* > 0.13) was found. The right dlPFC activation between groups during n-back is presented in Figure 3.

#### 3.2.2. vlPFC

Statistical comparisons with ANOVAs revealed no main effects or significant interactions for the comparison between acute BD patients vs. healthy controls (all *p* > 0.19) and remitted/A vs. remitted/R phase (all *p* > 0.25). However, remitted/R vs. controls revealed a significant main effect of group (F(1, 42) = 10.40, *p* = 0.002), showing that controls had a higher vlPFC activation. No other main effects or interactions were significant.

## 4. Discussion

Overall, the main hypotheses of the present study could be confirmed. The statistical main effects showed an increase in n-back reaction time, omission errors, and a trend towards lower frontal brain activation patterns in the acute phase of BD patients compared to controls. The reaction time of patients improved from the acute to the remitted state but could not reach the levels of healthy controls, especially in the more difficult conditions, which points to a more subtle cognitive deficit that can only be recognized in more difficult paradigms. Given the complexity of today’s working environment, this might be of relevance for the psychosocial functioning of remitted BD patients, underscoring the necessity of cognitive remediation programs. 

Nevertheless, it must be taken into account that improved performance could also be based on training effects. This improvement in reaction time is in line with previous work from our group [7] and was found in other research as well (Mann-Wrobel et al., 2011). The established finding of WM dysfunction in the BD acute phase [2] seems to conflict with studies where remitted BD patients, subtype I, were compared with healthy controls [9]. The latter review suggests the involvement of intact secondary systems in order to overcome the lack of integrity across WM circuits in BD patients, explaining why some remitted BD patients would have the same WM performance as controls [9]. In our study, an expected longer reaction time for remitted patients was found. 

In the present study, brain activation, localized to the right dorsolateral prefrontal area, was attenuated in acute patients compared to healthy controls. Moreover, remitted patients displayed, at least numerically, even stronger attenuation when compared to controls. It could be explained by the fact that remitted patients performed the task for the second time, and, thus, maybe recruited fewer neural resources due to learning effects. This would go in line with the observed shorter reaction times in the remitted phase compared to the acute phase. Similar effects of attenuated prefrontal brain activation elicited by a cognitive test with the same patients in the acute and the remitted phase of BD were reported [29]. An attenuation of brain activation has been shown before in other studies in acute patients [13,16,17,18,23,26] as well as in remitted patients [18,19,20,21,22,24,25,26,27,28], but this is the first study to show a change in prefrontal activation patterns during a working memory test over the course of the disorder. One possible explanation for this prefrontal attenuation for BD patients might be BD pathophysiology. Studies have hypothesized a dysregulation in frontolimbic connections [42] and a loss of homeostasis in important prefrontal–striatal–pallidal–thalamic–limbic brain networks [43], resulting in a disinhibition of temporal structures [42] and impairment in motor inhibition [43].

This study had a naturalistic design, as acute BD patients were medicated with mood stabilizers, antipsychotics, and antidepressant drugs during their inpatient stay. They remained on maintenance medication during their remitted state. Regardless of statistical effects, medication might have already been affecting brain activation while depressive symptomatology was still persisting, which could explain why acute patients showed physiologically increased activation patterns when compared to themselves in a remitted state. This effect was discussed in a review concluding that medication could normalize brain activity to the level of healthy controls [44]. When patients in this study were tested for the second time, i.e., in their remitted phase, their medication had changed. Thus, this frontal brain (i.e., dlPFC) attenuation and lack of WM are considered phenotypes of BD, and the medication for the acute phase might minimize this known physiological–functional pattern [45]. Since remitted subjects take less medication, it is possible to examine their brain activity and cognitive function in a more naturalistic state. 

Taken together, the literature on the n-back task and neuroimaging in BD patients suggests that, even with wide heterogeneity regarding accuracy and reaction time, BD patients in the remitted phase show irregular activation in the dorsolateral and ventrolateral PFC and impaired PFC network connectivity, which is related to WM [9]. However, little is known about the change in cognitive deficits as well as their underlying neurobiology in the course of BD patients’ phases. This is the first study to shed some light on the changes in prefrontal activation patterns during a working memory activity in BD patients over time. More research is needed in order to correctly place our findings into a picture of the development of working memory and its related brain activity over the course of BD and its different phases.

Although the total sample size of this study consists of 62 subjects, out of 32 patients, only 15 could be examined a second time, i.e., during the remitted phase. This sample size is rather small, as generally encountered in follow-up studies, due to various difficulties: some patients moved away, others were not remitted, and some patients just did not want to take part again. Since variance is high in imaging studies and effect sizes are small, it is possible that we missed differences in activation due to the sample size, especially for the contrast between acute and remitted patients. Patients were diagnosed with both bipolar I and bipolar II disorder, which are argued to be different disorders and, therefore, should optimally not be included in the same sample [43]. It would be recommendable to conduct future studies with larger samples in order to compare subgroups of BD patients with different subtypes or different disease loads.

Near-infrared spectroscopy is very well-tolerated by patients and controls and is much less expensive. However, one of the drawbacks of this method is the limited depth of the photons’ penetration to measure cerebral tissue oxygen perfusion. Near-infrared light can only be absorbed by blood flowing through cortical areas, and, therefore, only oxygenation changes in the cortex can actually be measured. Any changes happening in deeper, subcortical areas cannot be detected. As mentioned above, one of the major limitations of this study is that the n-back retest was not conducted in the control group. Therefore, the well-stablished learning effect could confound the observed behavioral and brain activity differences between the control group and the remitted/R phase BD patients, as well as between the BD phases (i.e., acute vs. remitted). One might hypothesize that the quite long interval between test and retest (i.e., at least three months) might limit the learning effect to a small extent. Moreover, differences in medication between the BD patients in the acute and remitted phase as well as inconsistent time intervals between test and retest might confound the results. Additionally, it is not known whether the patients had additional psychotherapy, whether in the past or ongoing. However, these are biases, which are commonly reported in studies investigating patients in such conditions and are hardly controllable.

## 5. Conclusions

Despite these caveats, our study contributes importantly to the growing body of research showing decreased working memory performance combined with attenuated patterns of activity in prefrontal brain activation in bipolar patients during cognitive tasks in the acute and remitted phase of disease. Future studies should account for the learning effect of the n-back test in order to better understand the change in cognitive deficits as well as their underlying neurobiology in the course of BD patients’ phases.

## Figures and Tables

**Figure 1 brainsci-13-00744-f001:**
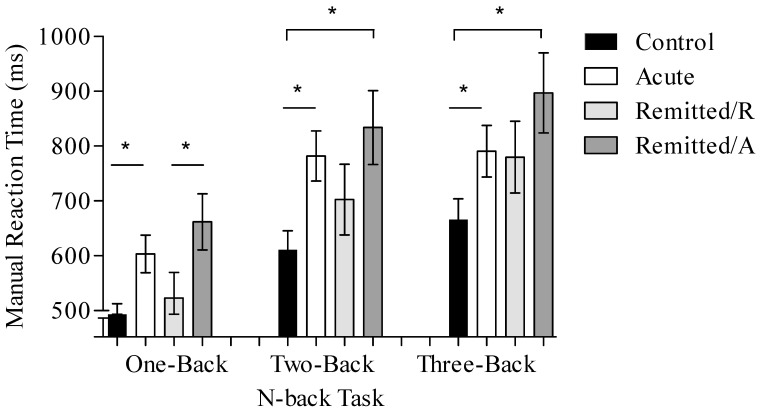
N-back task reaction times for acute BD patients (*n* = 32), remitted BD patients in their acute phase (remitted/A), and patients in their remitted phase (remitted/R; *n* = 15), as well as for healthy controls (*n* = 30) in the different task conditions (one-back, two-back, and three-back). Data are presented as mean values and bars indicate the standard error. An asterisk stands for significant difference between groups *p* ≤ 0.05.

**Figure 2 brainsci-13-00744-f002:**
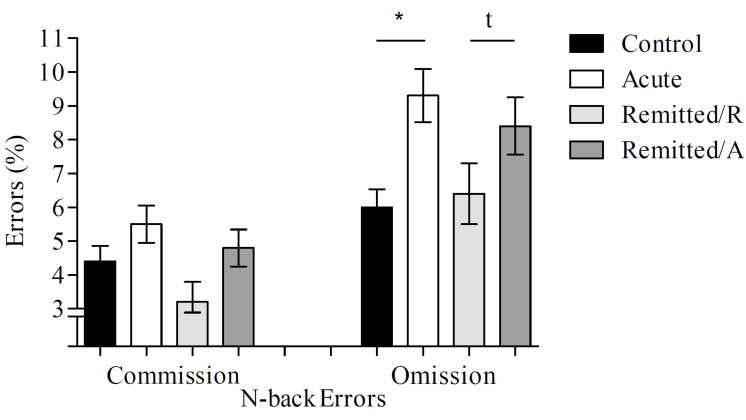
Commission and omission errors of acute BD patients (*n* = 32), remitted BD patients in their acute (remitted/A) and remitted (remitted/R) phases (*n* = 15) and healthy controls (*n* = 30). Error bars indicate the standard deviation, an asterisk indicates a significant difference (*p* ≤ 0.05), and a “t” indicates trend differences (*p* < 0.10) between groups/phases.

**Figure 3 brainsci-13-00744-f003:**
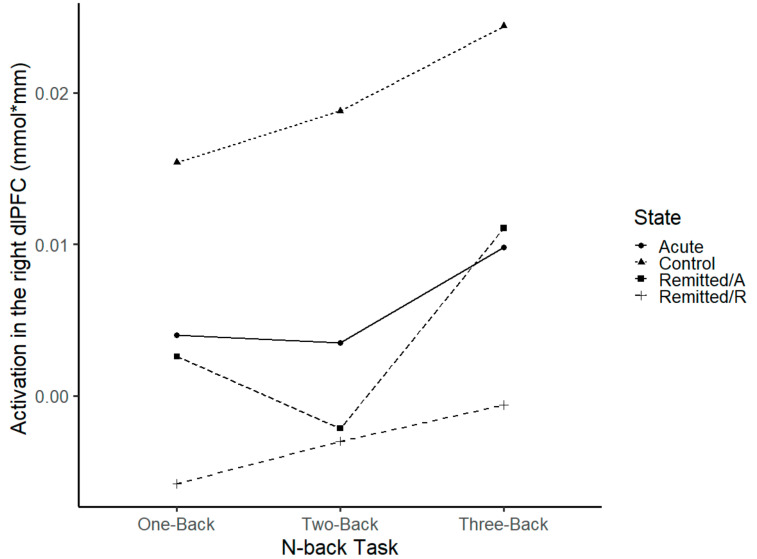
Brain oxygenation measured with fNIRS in the right dlPFC of acute BD patients (*n* = 31), remitted BD patients in their acute (remitted/A) and remitted (remitted/R) phases (*n* = 14), and healthy controls (*n* = 30) during n-back task conditions (one-back, two-back and three-back). Values are presented as mean and bars indicate the standard error. An asterisk indicates a significant difference (*p* ≤ 0.05) given for control compared to remitted/R group. A “t” indicates trend differences (*p* < 0.10) given in control vs. acute BD patients.

**Table 1 brainsci-13-00744-t001:** Demographic and working memory data of the sample.

	Healthy Controls (*n* = 30)	Acute BD (*n* = 32)	Remitted/A BD (*n* = 15)	Remitted/R BD (*n* = 15)
Sex ♂/♀ (*n*)	10/20	19/17	10/5	10/5
Age (years)	42.3 ± 10.7	42 ± 11.2	36 ± 11	36 ± 11
MADRS (score)	-	14.6 ± 10.3	18.2 ± 7.9	3.5 ± 1.5
YMRS (score)	-	3.4 ± 3.7	4 ± 3.5	1.1 ± 1.3
Working Memory				
One-Back (RT in ms)	485.81 ± 78.6	603.5 ± 146.3	661.89 ± 172.6	523.08 ± 103.2
Two-Back (RT in ms)	611.17 ± 119.9	782 ± 218.9	833.89 ± 264.5	702.48 ± 176.8
Three-Back (RT in ms)	665.91 ± 142	790.63 ± 237.5	896.88 ± 203.7	779.76 ± 200.1
Commission errors (%)	4.4 ± 3.5	5.5 ± 4.4	4.8 ± 4.1	3.2 ± 1.9
Omission errors (%)	6 ± 3.1	9.3 ± 4.7	8.4 ± 4.6	6.4 ± 2

Data are represented in mean ± SD. BD—bipolar disorder patients; Remitted/A—for remitted patients in acute phase; Remitted/R—for remitted patients in remitted phase; MADRS—Montgomery–Åsberg Depression Rating Scale; YMRS—Young Mania Rating Scale; RT—reaction time.

## Data Availability

The data presented in this study are available on request from the corresponding author. The data are not publicly available due to missing consent from test subjects.

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
