# Peer review of "Neural Responses to a Working Memory Task in Acute Depressed and Remitted Phases in Bipolar Patients"

_brainsci, 2023, doi:10.3390/brainsci13050744_

Round 1

Reviewer 1 Report

In this article, patients in the depressive phase of bipolar disorder were compared with healthy controls in terms of working memory and related brain activation, and then the available patients were re-evaluated in remission and re-examined in the same aspects.  The study will contribute to the studies in the literature in terms of its longitudinal design.  Results "decreased working memory performance in BD patients during working memory task in the acute phase of disease.  Working memory performance improved in the remitted phase of the disease, but was still attenuated in particular for the more demanding conditions" are generally similar to the literature. The hypothesis, methodology and sample of the study are appropriately conveyed, and limitations and strengths are expressed.  Although the sample size is small, it can serve as a reference for new studies.  The study needs some minor corrections;

1. The term acute period in the title may evoke a manic episode. For this reason, it should be stated in the title and appropriate places that the patients were in acute depressive period.

2. Although it was stated that the sample characteristics were included in another article, the characteristics of the bipolar process in the patients should be included in this article. In particular, findings such as the duration of the illness, when it started, and how long the treatment lasted should be included as they will affect the test results and brain activation. 

3. The dates of the study should be specified, this is necessary to explain why DSM-IV criteria were used in this study. 

4. Please give brief information about the OCPRIT diagnostic criteria, emphasizing how it differs from the DSM. 

5. The groups in which patients were reported to be depressed and recovered should be more clearly described. Explain what is used as a cut-off point for scale scores.

6. Specify whether the effects of age, gender, medications and their duration of use were controlled for. If this was not done, it should be included in the limitations.

7. The effects of being female, age range, use of psychotropic drugs on working memory, tests and brain activation should be addressed in the discussion. Other than this, the discussion

8. The conclusion section is expected to include different suggestions to guide new studies, please make the necessary editing. 

Author Response

Dear Reviewer,

We would like to kindly thank you for your remarks. We have considered your suggestions and would like to reply as followed:

  1. We have changed the title according to your specifications.
  2. We have added the description of the sample in the supplementary material.
  3. We have added the date of the study, which was 2010, in the methods  section.
  4. We have added additional information on the OPCRIT in the methods section.
  5. We have added additional information about the remitted sample group.
  6. That information can be found in the sample description which we have added to the supplementary material. We accounted for age and gender, but did not account for medication effects. We have added that to the limitations section.
  7. We have added a brief discussion about the effects of gender and medication.
  8. We have included different suggestions to guide new studies.

We thank you again for your review.

Reviewer 2 Report

I really appreciate the opportunity to review the manuscript brainsci-2312665 entitled:
"Neural responses to a working memory task in acute and remit-2 ted phases in bipolar patients"

I commend the authors for describing this critical and timely issue. The paper is interesting and well-written; however, I would like to highlight some issues that merit revision:

In the part where of describes the treatment received by the patients it appears well clear the pharmacological part, but it is not clear to the reader if the patients turn out to have some form of add-on psychotherapy. In view of the importance of such possible add-on therapy on Working Memory I beg the users to specify briefly in the text this aspect, or, if the data is not available to add it as a limitation to the manuscript

Author Response

Dear Reviewer 2,

we kindly thank you for your review of our paper. Since the status of the psychotherapy is not known for this sample, we have added that to the limitations section. We will incorporate this in our future studies.